

# Compound droughts under climate change in Switzerland

Christoph von Matt[1,2], Regula Muelchi[3], Lukas Gudmundsson[4], and Olivia Martius[1,2]

[1]Institute of Geography, University of Bern, Bern, Switzerland
[2]Oeschger Center for Climate Change Research, University of Bern, Bern, Switzerland
[3]Federal Office of Meteorology and Climatology MeteoSwiss, Zurich-Airport, Zurich, Switzerland
[4]Institute for Atmospheric and Climate Science, ETH Zurich, Zurich, Switzerland

**Correspondence:** Christoph von Matt (christoph.vonmatt@unibe.ch)

**Abstract.** The co-occurrence of meteorological, agricultural, and hydrological droughts (multivariate compound droughts) in Switzerland during growing season is problematic due to limitations in water abstractions from rivers during low flow periods, while at the same time the need for irrigation is high. We analyse compound droughts for 52 catchments in Switzerland during the extended summer season (May–Oct) using the transient climate and hydrological scenarios for Switzerland (CH2018 and Hydro-CH2018) for both a scenario with mitigation (RCP2.6, 8 model chains) and without mitigation (RCP8.5, 20 model chains). In the RCP8.5 scenario the number of compound drought days is projected to significantly increase by mid-century across all greater regions of Switzerland. The increased frequency is mainly a result of more frequent events (significant) rather than longer event durations (non-significant). Models generally agree on the sign of change. By 2085, compound drought events are projected to occur in median once per catchment per extended summer season north of the Alps and every 1-2 years south of the Alps. Further, the increases in compound drought days mainly occur between May–Oct leading to a shift into the main agricultural production season and a more pronounced seasonality with highest occurrence probabilities between mid-July and the begin of October. Coupled to the increase in days and events, significantly more catchments are projected to be affected by compound droughts at the same time. In the RCP2.6 (mitigation) scenario, the increase in the number of compound drought days and events is not significant by the end of the $21^{st}$ century. In comparison with RCP8.5, the number of compound drought days are reduced by 50-55% north of the Alps and up to 75% south of the Alps by the end of the century. This emphasizes the need for coordinated adaptation in combination with mitigation measures taken at early stage.

## 1 Introduction

Droughts are a globally recurring phenomenon of the natural hydrological cycle and describe a sustained period of below normal water availability (Tallaksen and Van Lanen, 2004). In recent years, Switzerland experienced several drought events, most notably in the years 2003, 2011, 2015, 2018, 2020, and 2022 (BAFU et al. (Hrsg.), 2019; BAFU, 2016; Brunner et al., 2019c; BUWAL, BWG, MeteoSchweiz, 2004; Scherrer et al., 2022). These droughts led to severe ecological and economic impacts (BAFU et al. (Hrsg.), 2019; Keller and Fuhrer, 2004). In 2022, combined heat and drought conditions were in fact unprecedented in most parts of Europe (Tripathy and Mishra, 2023). Depending on the characteristics of a drought (e.g., type, severity, magnitude, timing and duration, spatial extent) various sectors/players are affected such as agriculture (e.g., yield



losses, reduced milk production), energy (e.g., hydropower generation) or the public (e.g., water supply, air quality) associated with substantial economic losses (Ranasinghe et al., 2021; Haile et al., 2020; BAFU (Hrsg.), 2021; Hendrawan et al., 2023; Otero et al., 2023; Naumann et al., 2021).

There is no single definition of droughts that covers all aspects of droughts (Wilwhite and Glantz, 1985; Lloyd-Hughes, 2014; Van Loon, 2015; Brunner et al., 2021; Ault, 2020). Droughts are often classified into 1) meteorological droughts that are re-

lated to a precipitation deficit, 2) agricultural or soil moisture droughts that are related to soil moisture deficits, 3) hydrological droughts that are often defined as streamflow deficit, and 4) socio-economic droughts that include impacts resulting from all drought types (Wilwhite and Glantz, 1985; Haile et al., 2020; Van Loon, 2015; Mishra and Singh, 2010; Savelli et al., 2022). Droughts are commonly identified using drought indices which are derived from drought indicators (e.g., precipitation deficits, low soil moisture, low streamflow) (Mukherjee et al., 2018; Faiz et al., 2021; Van Loon, 2015; Yihdego et al., 2019; Bachmair

et al., 2016). Most drought indices are representative for one specific drought type, for example the Standardized Precipitation Index (SPI, McKee et al., 1993) captures meterological droughts, the Soil Moisture Anomaly (SMA, Orlowsky and Seneviratne (2013)) captures soil moisture drought and the Standardized Runoff Anomaly (SRA, Gudmundsson and Seneviratne (2015b)) describes hydrological droughts. However droughts indices can also serve as proxy for several drought types depending on aggregation times (Cammalleri et al., 2019; Haslinger et al., 2014). Standardized indices such as the SPI have become popular

in recent years and are now widely implemented in drought monitoring, drought prediction, and drought early warning systems (DEWS, see e.g.,  Bachmair et al., 2016; Kchouk et al., 2022; Tijdeman et al., 2020)). In addition, index combinations or combined multivariate drought indices (e.g., the Combined Drought Indicator (CDI) at JRC) exist (Bachmair et al., 2016; Cammalleri et al., 2021). However, no single drought index can capture all aspects of a drought and perform best in all situations, regions, or climates, which makes the index selection process a site-specific task (Bachmair et al., 2016, 2018; Hayes

et al., 2011; Myronidis et al., 2018; WMO and GWP, 2016; Hall and Leng, 2019; Van Loon, 2015).

Droughts typically emerge from a period of anomalously low precipitation (*meteorological drought*). If such a period persists or co-occurs with a period of high evaporative demand, soil moisture storages may deplete resulting in a soil moisture deficit (*soil moisture* or *agricultural drought*) and eventually in plant-water stress (Floriancic et al., 2020; Mishra and Singh,

2010; Seneviratne, 2012; Van Loon, 2015; Zhao et al., 2022). The drought signal may ultimately propagate into the hydrological system causing a streamflow or even a groundwater deficit (*hydrological* or *groundwater drought*) (Haile et al., 2020; Van Loon, 2015; Mishra and Singh, 2010; Van Loon and Van Lanen, 2012). The exact sequence of the drought signal translation through the hydro-terrestrial system may differ depending on drought typology, drought generating processes, and on human interactions (e.g., water abstractions) and is strongly non-linear (Brunner et al., 2023; Haile et al., 2020; Savelli et al.,

2022; Tijdeman et al., 2018; Van Loon, 2015; Van Loon and Van Lanen, 2012). While meteorological droughts are strongly tied to climate variability, soil moisture and hydrological drought characteristics are spatio-temporally more variable due to the importance of local factors such as water storage and release or catchment characteristics (e.g., Apurv et al., 2017; Apurv and Cai, 2020; Denissen et al., 2020; Haslinger et al., 2014; Peña-Angulo et al., 2022; Staudinger et al., 2017, 2014; Sutanto and





Van Lanen, 2022; Tijdeman et al., 2018; Van Lanen et al., 2013)


While assessing past drought trends has so far often been hampered by the length and availability of historical records (Brunner et al., 2021; Hasan et al., 2019; Sheffield et al., 2012; Vicente-Serrano et al., 2022; Kohn et al., 2019), many long-term drought drivers changed in an unfavourable direction (de Jager et al., 2022). Further, many studies project significant future increases in drought hazards, risk and impacts in many regions around the world (Arias et al., 2021; Forzieri et al., 2014;

Grillakis, 2019; Gudmundsson and Seneviratne, 2016; Lehner et al., 2017; Naumann et al., 2021; Spinoni et al., 2020, 2018; Trenberth et al., 2014; Zeng et al., 2022). This goes along with an intensification of the water cycle and increased evaporative demand under global warming (Arias et al., 2021).

In Switzerland, future changes in annual precipitation and annual streamflow discharge are subject to uncertainties with projections indicating only minor changes for both (Kotlarski et al., 2023; CH2018, 2018; BAFU (Hrsg.), 2021; Muelchi et al.,

2021b; Brunner et al., 2019b). This is partly due to the proximity to the Alps, which act both as "water tower" and "Alpine Divide" (Viviroli et al., 2007; van Tiel et al., 2023; Haslinger et al., 2019) placing Switzerland in a transition zone between projected (winter) wetting trends in northern Europe and drying trends in southern Europe (Spinoni et al., 2018; Grillakis, 2019; Hirschi et al., 2020; Gudmundsson et al., 2017; Gudmundsson and Seneviratne, 2015a; CH2018, 2018; Forzieri et al., 2014). Changes are however more pronounced on a seasonal scale with significant projected drying trends in summer without

migiation efforts for both meteorological and agricultural drought indices (CH2018, 2018; Fischer et al., 2015), minimum discharges (Brunner et al., 2019b), as well as moderate low flow extremes (Muelchi et al., 2021a). Without mitigation, (meteorological) dry periods are projected to get longer, more frequent, and more severe (CH2018, 2018; Hirschi et al., 2020; Spinoni et al., 2018; Kotlarski et al., 2023). Along with rising temperatures, also the evaporative demand is projected to increase, leading to a augmented need for irrigation in many important agricultural regions of Switzerland (Allgaier Leuch et al., 2017;

Fuhrer and Calanca, 2014; Hirschi et al., 2020; Holzkämper et al., 2020; Lanz, 2020; Remund et al., 2016; Vicente-Serrano et al., 2022). Water scarcity is currently not an issue in Switzerland in large catchments, however, some lower lying small-to mid-size catchments in important agricultural regions of Switzerland do already face problems in extremely dry summers (BAFU et al. (Hrsg.), 2019; BAFU (Hrsg.), 2021; Brunner et al., 2019a). Extreme summers like in the years 2003 and 2022 will become more average under climate change projections in future (Calanca, 2007; Imfeld et al., 2022b, a; Miralles et al.,

2019). As consequence, water scarcity is projected to further aggravate in regions where increased water demand may not be fully compensated by current adaptive capacities (e.g., natural and artificial reservoirs) (Fuhrer and Calanca, 2014; Brunner et al., 2019a; Henne et al., 2018; Lanz, 2020).

Drought impacts may become particularly severe if meteorological, agricultural, and hydrological droughts occur simultane-

ously, forming compound drought events (Markonis et al., 2021; Wu et al., 2022). Compound events describe situations where multiple hazards or drivers of hazards occur simultaneously or consecutively, thereby contributing jointly to societal or environmental risk (Zscheischler et al., 2020; Seneviratne et al., 2012). In case of compound drought events and following drought propagation principles, an extended period of precipitation deficit (meteorological drought) may lead to the propagation of the





drought signal to a soil moisture deficit critical for plant-water stress resulting in an augmented need for (agricultural) irriga-
tion. Irrigation may, however, not be possible due to (surface) water abstraction restrictions imposed during a simultaneous
hydrological drought as was the case in Switzerland for example in 2018 (BAFU et al. (Hrsg.), 2019). Further, compound
drought events can affect multiple mutually dependent regions or catchments at the same time, which can then pose additional
challenges to water management and may aggravate drought-related impacts (Singh et al., 2021). In case of such spatially
extended compound drought events, local adaptation measures alone may not be sufficient anymore to address them (Brunner
et al., 2019a; Fuhrer and Calanca, 2014; Singh et al., 2021; Tellman and Eakin, 2022; Kreibich et al., 2022; Kruse and Seidl,
2013). Studies investigating compound drought events often focus on compounding hot and dry extremes (Dirmeyer et al.,
2021; Manning et al., 2019; Otero et al., 2023), on temporal aspects (Brunner and Stahl, 2023; Li et al., 2022), or on spatial
concurrence (Brunner and Gilleland, 2021; Singh et al., 2021). Multiple drought types are more often considered in drought
propagation contexts (Ding et al., 2021; Gu et al., 2020; Jiang et al., 2023; Tijdeman et al., 2022; Van Loon and Van Lanen,
2012; Wu et al., 2021) and few studies quantify co-occurrence in terms of compound drought events, thereby often focusing
on two drought types (Brunner et al., 2019c; Manning et al., 2018; Wu et al., 2022).

In this work, compound drought events – defined as the simultaneous occurrence of meteorological, agricultural, and hy-
drological droughts - are investigated in 52 catchments in Switzerland for both present and future climates using the transient,
downscaled climate and hydrological scenarios for Switzerland (CH2018, Hydro-CH2018) (CH2018, 2018; Muelchi et al.,
2022). The main goal is to assess present and future characteristics of compound drought events in Switzerland to provide
crucial information for future water resources management, adaptation planning and motivation for mitigation actions.
The following questions are addressed: 1) How often do compound droughts, defined as the simultaneous occurrence of meteo-
rological, agricultural, and hydrological drought occur in Swiss catchments? 2) How are they characterised in terms of number
of days, duration, frequency and seasonality and how do these characteristics change in the future? 3) Are there regional dif-
ferences in compound drought occurrences and their characteristics? 4) How are compound droughts characterised in terms
of spatial extent? Do compound droughts occur simultaneously over multiple catchments or regions in Switzerland (*spatially
compounding droughts*)?
The remainder of this paper is structured as follows: 2 Data, 3 Methods, 4 Multivariate compound droughts, 5 Spatial extent
of multivariate compound droughts (*spatially compounding droughts*), 6 Discussion and 7 Conclusions.

## 2 Data

Compound droughts occurring during the extended summer season from May to October are analysed for 52 small- to mid-size
(17–1702 km2), lower lying catchments (mean altitude <1500m asl) in Switzerland. The catchments are distributed across the
greater (natural) regions of Switzerland defined by the FOEN (NCCS, 2023): Pre-Alps (n=22), the Jura (n=13), the Southern
Alps (n=4), and the Swiss Plateau Region (n=13). The locations of the catchments, the catchment outlets, and the catchment
affiliations to the Greater regions are shown in Fig. 1. The restriction to lower lying small- to mid-size catchments and to the

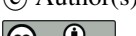



extended summer season ensures a focus on classical rainfall deficit droughts and limited influence of snow- and glacier-melt on streamflow discharge (Brunner et al., 2019b; Floriancic et al., 2020; Muelchi et al., 2021a, b).

The following model variables are used (daily means): precipitation (P), actual (ET), potential evapo(transpi)ration (PET), and
runoff (Q). Daily precipitation is taken from the gridded Swiss climate scenarios CH2018 (DAILY-GRIDDED, CH2018 Project Team (2018); Kotlarski and Rajczak (2018)) and aggregated over the catchment area. The scenarios consist of transient daily simulations for the period 1981–2099 that were statistically downscaled and bias corrected from EURO-CORDEX simulations (Jacob et al., 2014; Kotlarski et al., 2014; Kotlarski and Rajczak, 2018). For daily ET, PET, and Q, we use the Hydro-CH2018-Runoff ensemble simulations (Muelchi et al., 2022), which were driven by temperature and precipitation data from the CH2018
scenarios. This ensures temporal consistency between the meteorological and hydrological variables. The hydrological simulations are run with the semi-distributed hydrological model PREVAH (Viviroli et al., 2009) which accounts for processes such as evapotranspiration, soil moisture, and snow processes. PET is calculated by the Hamon equations (Hamon-PET; Hamon (1961)) and ET consists of both evaporation terms from both interception and soil moisture storages (see Viviroli et al., 2009). PREVAH was calibrated such that also low flows are represented satisfactorily (inverted flow calibration; Muelchi et al.
140 (2022)).

All simulations are available for the following representative concentration pathways (RCPs): RCP2.6 represents a mitigation-scenario and RCP8.5 a non-mitigation scenario (Moss et al., 2010; van Vuuren et al., 2011). In this study, we use the same model selection as described in Muelchi et al. (2021a) (see *Table 1* therein), which consists in total of 28 GCM-RCM model chains (RCP2.6: 8; RCP8.5: 20) and an observation-driven validation simulation (hereafter *CTRL*). This CTRL simulation
was driven by daily gridded precipitation (RhiresD) and temperature (TabsD) data provided by MeteoSwiss (see MeteoSwiss, 2021a, b; Frei and Schär, 1998; Frei, 2014) , which were also used for the bias correction of the CH2018 scenarios. For more information see Muelchi et al. (2022).

## 3 Methods

### 3.1 Drought indices, drought events and event characteristics

Each drought type is represented by an individual drought index. Drought events are defined based on a fixed-threshold approach and event characteristics are extracted by run theory principles (Yevjevich, 1967). We focus mainly on the characteristics *event duration*, *event frequency* or *sample return period*, *event seasonality* and *spatial extent*. The *event duration* corresponds to the time span during which index values fall below the fixed threshold. The *event frequency* corresponds to the number of
events occurring per a predefined period (e.g., extended summer season). The *return period* is derived from inversion of the empirical event frequencies (e.g., 1/frequency). *Event seasonality* refers to the day of the year (DOY) on which a specific drought day occurs. The *spatial extent* is quantified by the number of catchments simultaneously affected by drought conditions. To account for statistical independence of consecutive and potentially dependent drought events, an additional inter-event distance pooling procedure is applied. If the time between two consecutive drought events is smaller than the inter-event distance (in


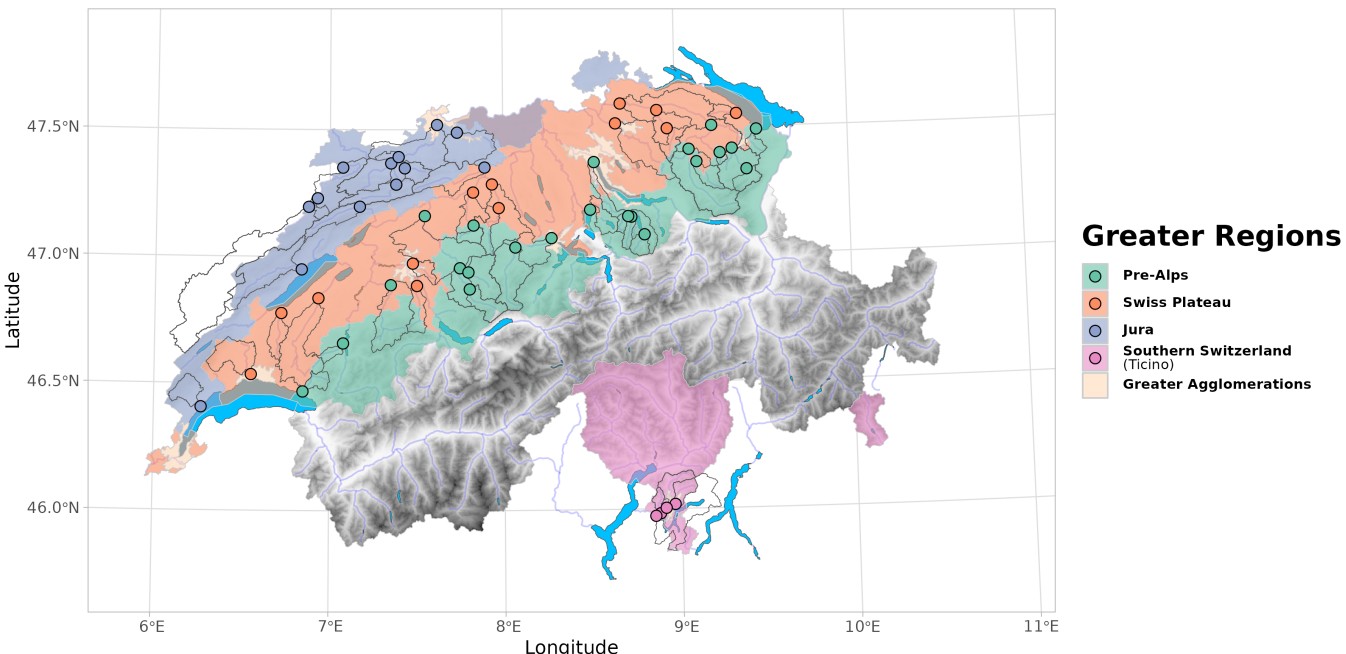

**Figure 1.** Overview of study region: the 52 analysed catchments with a mean altitude <1500 m asl (countours and outlet points) and their classification into four Greater Regions in Switzerland.

days), the two drought events are pooled and considered as one single event (see e.g., Sarailidis et al., 2019; Tallaksen et al., 1997). Indices and thresholds for event extraction are defined based on the chosen reference period 1991-2020. For more details see the corresponding sections on individual drought types.

### 3.1.1 Meteorological Drought - Standardized Precipitation Index (SPI-3)

To represent meteorological drought we use the Standardized Precipitation Index (SPI, McKee et al. (1993)) on a three-monthly scale (SPI-3). The SPI is a widely used meteorological drought index based only on precipitation and is also recommended by the World Meteorological Organization (WMO and GWP, 2016; Bachmair et al., 2016; Hayes et al., 2011). For each catchment time series, the retrospective three-monthly (aggregated) precipitation data is fitted to a Gamma distribution first and then transformed to a standard normal distribution (as recommended by Stagge et al. (2015)). To calculate daily SPI-values, the *SCI* R-package (Gudmundsson and Stagge, 2016; Stagge et al., 2015) was used and functions were adapted for daily temporal resolution. Following Stagge et al. (2015) all years of the time series were homogenized to 365 days by excluding the last day of the year in case of leap years. Gamma fits were derived separately for each day of the year for the reference period 1991-2020. In case of non-convergence of the Gamma distribution fit, missing daily SPI-values were interpolated by averaging the two adjacent SPI values. In this work, we focus on moderate SPI events which are commonly defined by a fixed threshold of $SPI < -1$ (Tschurr et al., 2020; McKee et al., 1993). Extracted SPI events are pooled by an inter-event distance of 7 days.





### 3.1.2 Agricultural Drought - Relative Evapotranspiration (ET/PET)


Soil moisture drought or agricultural drought represents the next stage in the concept of drought propagation of classical rainfall-deficit droughts (e.g., Seneviratne, 2012; Van Loon and Van Lanen, 2012). We aim to capture situations indicative of plant water stress that necessitate irrigation to avoid potential yield losses. A drought index frequently used to indicate plant water stress and irrigation need is the relative evapotranspiration (ET/PET), which corresponds to the ratio of actual (ET) to potential evapo(transpiration) (PET) (e.g., Fu et al., 2022a; Fuhrer and Jasper, 2009; Otkin et al., 2018; Ranasinghe et al., 2021;

Remund et al., 2016; Seneviratne et al., 2010; Walthert et al., 2015). We use ET/PET values averaged over three days. Studies investigating agricultural droughts based on ET/PET often use a threshold of $ET/PET < 0.8$ to identify situations with plant water stress (Remund et al., 2016; Allgaier Leuch et al., 2017; Walthert et al., 2015; Fuhrer and Jasper, 2009; Sawadogo et al., 2020). The threshold was (also analytically) derived for forest vegetation in Switzerland by Walthert et al. (2015).

Plant responses to ET/PET may, however, differ among hydro-climatic different regions due to localized plant-evolutionary adaptations, which can complicate the choice of an adequate fixed threshold (Denissen et al., 2020; Fu et al., 2022a, b; Stocker et al., 2023; Miralles et al., 2019). By using a fixed threshold, this study focuses on hydro-climatic differences in agricultural drought conditions rather than local plant-specific extreme conditions. Like Fuhrer and Jasper (2009) an inter-event distance pooling of two days conditional on five days event duration is applied.

### 3.1.3 Hydrological Drought - seven day average streamflow (M7Q / M7)

To represent hydrological drought, the 7-day average streamflow (M7Q) is used. M7Q also serves as basis for low flow indices analysed by the Federal Office for the Environment (see e.g., BAFU, 2019; Kohn et al., 2019). In Switzerland, the water protection law Art. 4 specifies a residual flow ("Restwassermenge") named "Q347" that must be retained after water abstractions (BUWAL, 1997; Kohn et al., 2019; Weingartner and Schwanbeck, 2020; Tallaksen and Van Lanen, 2004; Aschwanden and

Kan, 1999; Aschwanden, 1992). During hydrological drought situations with extreme low flow conditions, restrictions based on Q347 may be imposed inhibiting water abstraction for agricultural irrigation as happened for example in 2018 (BAFU et al. (Hrsg.), 2019). We define hydrological droughts similarly by using the $5^{th}$ percentile of M7Q over the entire year and the full 30-year reference period based on the flow duration curve (*hydroTSM* R-package, Zambrano-Bigiarini, 2020). Like in Van Loon and Van Lanen (2012), a combination of inter-event distance pooling and minimum event duration is used. Here,

events are pooled using a seven day inter-event distance and only events with a minimum duration of 10 days are considered.

### 3.2 Compound drought types and their analysis

Compound events can be classified in preconditioned, multivariate, temporally and spatially compounding events (Zscheischler et al., 2020). Here, two types of compound events are investigated, namely multivariate and spatially compounding events.





### 3.2.1 Multivariate compound droughts

Multivariate compound droughts are defined here as the simultaneous occurrence of multiple drought types (meteorological, agricultural, hydrological) within the same catchment. Based on the drought events of meteorological, agricultural and hydrological droughts (see red phases in the time series of SPI-3, ET/PET and M7Q in Fig. 2), the number of drought types simultaneously in drought phase is counted. Compound drought events occur if at least two different drought types occur simultaneously (yellow and red phases in bottom row of Fig. 2). Here we focus only on compound droughts of all three drought types (red phases). Compound droughts are analysed in terms of the number of days (count-based), event frequency and duration, and seasonality. Compound drought events correspond to contiguous days with compound drought conditions (red phases in bottom row of Fig. 2). The event duration corresponds to the number of contiguous compound drought days of a specific event. Compound drought events were extracted analogous to individual drought types (see Section 3.1) but no further pooling was applied as event independence was addressed for individual drought types. The seasonality was quantified by the frequency of compound drought days for each day of the year (DOY) and subsequent estimation of (Gaussian kernel) density distributions with the $tidyverse$-package (Wickham et al., 2019). Except for seasonality, all characteristics are assessed only for events and days occurring within the extended summer season (May–Oct). Events with at least one drought day occurring within the period May–Oct were considered in the analysis. If not stated otherwise the number of days and the event frequency is *per extended summer season*).

Note that the spatial extent of compound droughts is introduced as separate compound event type in section 3.2.2.

To summarise projected changes in compound drought event characteristics under climate change scenarios, catchments were aggregated into Greater regions. An aggregation based on Greater regions is equivalent to a simplified grouping by catchments with similar runoff regime types (see e.g., Weingartner and Schwanbeck, 2020; Muelchi et al., 2021b; Brunner et al., 2019b; Floriancic et al., 2020; Kohn et al., 2019; Aschwanden and Weingartner, 1985). Classification is based on percentage overlap of catchment area and the Greater regions and was manually checked for consistency. The catchment selection and grouping to the Greater regions are shown in Figure 1.

The quality of the meteorological and hydrological data has been previously validated for dry period lengths, SPI, simulated runoffs, and moderate runoff extremes (see e.g., Rajczak et al., 2016; Tschurr et al., 2020; Muelchi et al., 2022, 2021a). We use the observation-driven hydrological validation simulation (*CTRL*-run) to contextualize and validate our results (Muelchi et al., 2022). Characteristics and statistics were compared for the period 1991-2014, which is the longest available period overlapping with the reference period of scenario-driven simulations. The results of the validation are presented in the *Supplementary Material*.

### 3.2.2 Spatial extent of the compound droughts *(Spatially compounding events)*

The spatial extent of compound drought events (or *spatially compounding events*) is quantified by the number of catchments simultaneously affected by multivariate compound drought conditions over all 52 analysed catchments independent of Greater regions.


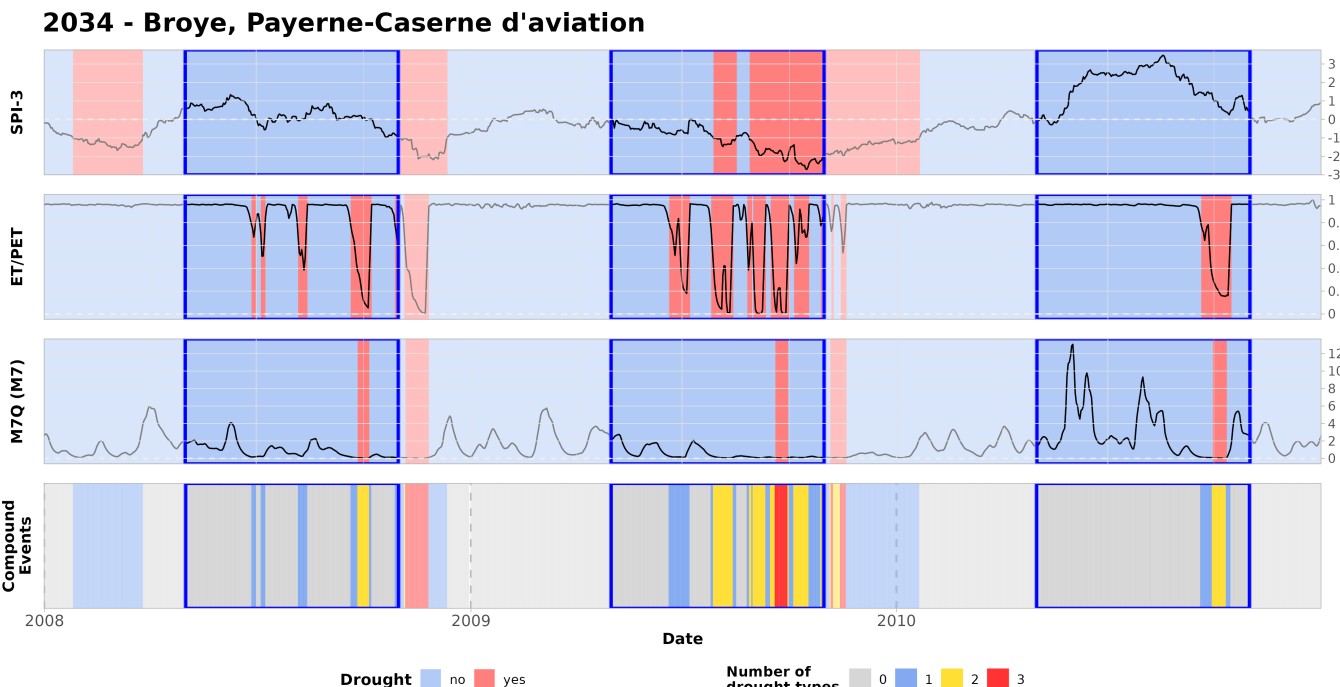

**Figure 2.** Example time series for catchment 2034 - Broye, Payerne - Caserne d'aviation (Swiss Plateau region). **(a)** Indicator time series for meteorological droughts represented by SPI-3, units indicate standard deviations (top row), agricultural drought by ET/PET, units indicate the fraction (second row), and hydrological drought by M7Q units indicate the discharge in $mmd^{-1}$ (third row). Drought phases are indicated in red (left legend). The blue rectangles indicate the extended summer season. **(b)** The number of drought types that are simultaneously in a drought phase are indicated by color. Note the different color scheme (right legend). Multivariate compound droughts with all drought types are marked in red, two concurrent drought types are indicated in yellow, one active drought type is indicated in blue, no active drought type is indicated in grey.

For each compound drought day, the number of catchments simultaneously affected by multivariate compound drought conditions (all three drought types, red phases) was determined for each climate model chain (see example time series in Fig. 3). We then analysed the median number of simultaneously affected catchments per compound drought day and cumulative distributions of days conditional on the number of simultaneously affected catchments. For the analysis only days within the extended summer season were considered. The results for individual models were then aggregated per emission scenario (RCP2.6 / RCP8.5).

For the event-based analysis, events were then extracted based on the time series of the number of simultaneously affected catchments (hence the separate event definition). Events are characterised by the number of events, the event duration, and the maximum number of simultaneously affected catchments over the entire duration of a specific event. The event duration spans over contiguous days with at least one catchment affected by compound drought conditions and ends if no single catchment





shows compound drought conditions. As for multivariate compound drought events, only events within the extended summer season were considered and no pooling was conducted.

The timing of the maximum number of catchments concurrently experiencing compound drought conditions was not assessed
and may occur at any time during an event. Further, catchments affected simultaneously by compounding drought events can be located anywhere in Switzerland (see individual catchment time series in Fig. 3) and we did not require spatial proximity or catchment connectivity.

Note for the interpretation of our results that spatially compounding events do technically not occur until at least two catchments are simultaneously affected by compound droughts events. For the sake of clarity, we hereafter mainly refer to the spatial
extent of compound droughts.

## 3.3 Assessment of the projected changes under climate change

Projected changes in compound drought characteristics related to climate change were assessed by comparing distributions over 30-year periods commonly used in climate change and climate impact assessments, namely 2035 (2020-2049 or *near future*), 2060 (2045-2074 or *mid-century*) and 2085 (2070-2099 or *end-of-century*) with the reference period 1991-2020 (WMO,
2007; CH2018, 2018; Muelchi et al., 2021a). Note that the reference period was shifted compared to other studies (e.g., Muelchi et al., 2021a) to avoid inconsistencies in soil moisture resulting from the spin-up process of the hydrological model (not shown) and include some years (ca. 10 years; see e.g., van Vuuren et al., 2011) that are forced with scenarios.

For all characteristics, multi-model median distributions are analysed considering all model chains per RCP-scenario. Significance in differences of the median among all models is then assessed by strict non-overlapping confidence intervals (CI) of the
median derived by the formula $\pm 1.58 \times \frac{IQR}{\sqrt{n}}$, where n denotes the sample size (McGill et al., 1978).

To quantify uncertainty and robustness of changes of the selected ensemble, an approach similar as in IPCC AR6 is used, which quantifies significance of changes in multi-model medians in combination with indications on climate model agreement concerning the sign of change (CH2018, 2018; Lee et al., 2021). Changes are considered likely when at least 90% of model chains per RCP agree on the sign of the change, which hereafter is referred to as *general model agreement*. Due to the potential
effects of the forcing scenarios and due to the difference in sample size (RCP2.6 8 model chains, RCP8.5 20 model chains), the reference period was assessed separately for each scenario.

An exception exists for the analysis of the relationship between the duration and maximum number of simultaneously affected catchments (see Section 5.2), where events of all models were pooled per scenario and no differentiation among periods were made to increase sample sizes for more rare events affecting large parts of the country. In this case, significance was assessed
for differences among scenario rather than to the reference period.

**Figure 3.** The spatial extent of compound drought events over Switzerland. The figure shows exemplary RCP8.5 scenario model simulation by CLMCOM-CCLM5-ECEARTH EUR-44km. **(a)** Time series of the number of simultaneously active drought types (color-shading) for all 52 catchments. Each row corresponds to one catchments and the catchments are sorted by Greater regions and longitude (top = East, bottom = West) for 1991-2099. Multivariate compound drought days highlighted in red (all three drought types active). **(b)** Time series of the number of catchments concurrently affected by multivariate compound drought conditions (red phases). **(c)** Zoomed in view on the time series (b) for the year 2018. Blue stripes indicate extracted spatially compounding events.

# 4 Multivariate compound droughts

## 4.1 Compound drought days

The number of compound drought days and its percent change is depicted in Fig. 4 for different periods and RCP2.6 and RCP8.5 forcing. In the reference period the median number of compound drought days is 3–6 days per catchment and per extended summer season in Switzerland. Most compound drought days occur in the Jura and Swiss Plateau region with ap-



proximately 6 days per extended summer season in median across all catchments. Fewer compound drought days per extended summer season occur in Southern Switzerland (approx. 4 days) and in the Pre-Alps (between 3–4 days).

In near future (2035), no significant changes are projected for the Greater regions for RCP2.6 and RCP8.5 (see Fig. 4b,f) and absolute numbers of compound drought days are comparable among Greater regions and scenarios. Under RCP2.6, the
number of compound drought days is slightly but not significantly increasing under climate change. Inter-regional differences remain constant by the end of the century under RCP2.6 with the Jura region still experiencing most compound drought days with 9 days per extended summer season in multi-model median followed by the Swiss Plateau region with 8.5 days and with 7.2 days slightly fewer in the Pre-Alps. Southern Switzerland is projected to experience 4.5 compound drought days in multi-model median (Fig. 4d). Further, ranges in model median distributions increase by the end of the century indicating increasing
uncertainty in possible future changes north of the Alps (Jura, Swiss Plateau and Pre-Alps), while they decrease in southern Switzerland (not shown). Highest model agreement is projected for the Pre-Alps and Southern Switzerland with 87.5% (7/8) models agreeing on more compound drought days while changes are more uncertain for the Swiss Plateau (75% (6/8)) and Jura region (62.5% (5/8)). Overall, the multi-model percental increases correspond to less than a doubling of number of compound drought days in all Greater regions (see Fig. 4i–k).


Without mitigation (RCP8.5), significant increases in compound drought days are projected for all Greater regions by mid-century and models do generally agree on an increase north of the Alps (Jura, Swiss Plateau and Pre-Alps) and by the end of the century also in southern Switzerland (Ticino) (Fig. 4g,h). The highest number of compound drought days are projected to occur in the Jura region with a multi-model median of 12.6 days per catchment and per extended summer season by 2060 and
19.4 days by 2085. The second highest number of compound droughts days are projected to occur in the Swiss Plateau region with 11.4 days (2060) and 18 days (2085) per extended summer season (Fig. 4g,h). By 2085 the number of compound drought days in southern Switzerland is similar to that of the the Swiss Plateau region with a (multi-model) median of 17.6 compound drought days, while the Pre-Alps experience fewer days with 15.8 days (Fig. 4h). Overall, the changes in median of absolute numbers of compound drought days are similar for all Greater regions, leading to smaller relative differences between regions.
The projected increases under RCP8.5 corresponds to 3.8-fold (multi-model) increase in the Jura region, a 4.5-fold increase in southern Switzerland and a 5-fold increase in the Swiss Plateau region and a 6.5-fold increase in the Pre-Alps by the end of the century (Fig. 4n). By mid-century, model agreement in the Jura and Pre-Alps reaches 95% (19/20 models agree). Most models do however also agree on an increase in southern Switzerland (75% or 15/20 models). By 2085 almost all models project an increase in the median number of compound drought days compared to the reference period in all Greater regions (N-CH:
100%, S-CH: 95%) and compared to the mid 21$^{st}$ century the increase appears accelerated.

## 4.2 Compound drought event duration

In the reference period, events last 11 days in the Jura and the Swiss Plateau regions and 9 days in the Pre-Alps, which is consistent with the regional differences in univariate hydrological droughts (Fig. 5a,e). Events last longest in Southern Switzerland with a median event duration of 16 days (Fig. 5a,e). The median event duration increases by the end of the 21st




## Compound drought days

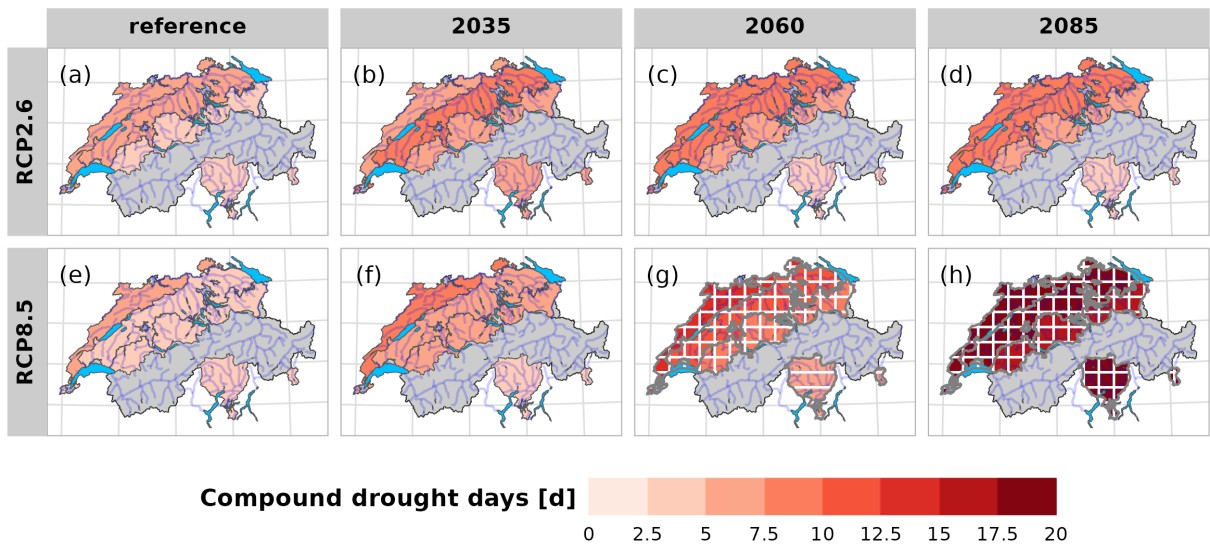

## Percentage change

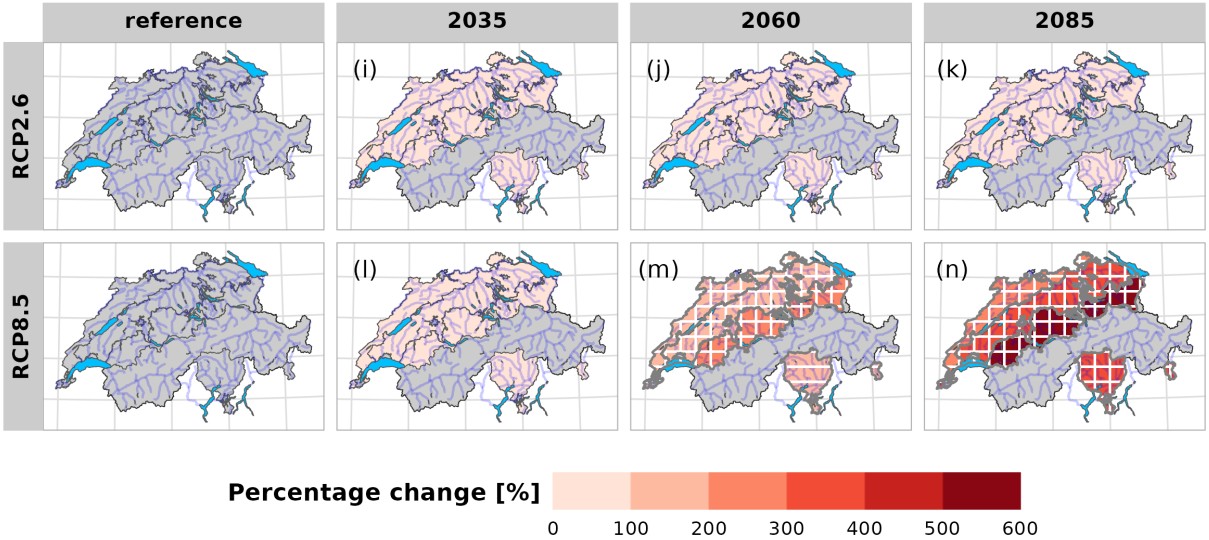

**Figure 4.** Compound drought days per extended summer season for the reference climate and under climate change for a mitigation (RCP2.6) and non-mitigation (RCP8.5) scenario aggregated on the Greater regions (median over all catchments). Presented are both the median value (top) and percentage change (bottom) over all model simulations per scenario. Hatching indicates if changes are significant compared to the reference period (horizontal lines) and whether ≥ 90% of model simulations agree on the projected changes (per scenario; vertical lines). Significance and model agreement were evaluated only for absolute values and are identical for both median values and percentage changes.





century under RCP8.5 (Fig. 5). The increase is, however, neither significant nor do most models agree on an increase with the exception of the Jura region for 2060, see Fig. 5g). By 2085 and without mitigation projected non-significant median increase in duration is stronger in regions with longer event durations (+21% for Jura and Swiss Plateau, +26% in Southern Switzerland and +11% in the Pre-Alps; see Fig. 5n) with longest events occurring in Southern Switzerland (19 days), followed by Jura and Swiss Plateau region (13 days) and the Pre-Alps (10 days) (Fig. 5h).

## 4.3 Compound drought event frequency (sample return period)

Compound drought events occur in (multi-model) median every 2.1 years in the Jura, every 2.5 years in the Swiss Plateau region and every 3.2 years in the Pre-Alps in the reference period. In Southern Switzerland an event occurs in median every 4.4 years (Fig. 6a,e). Projected changes in event frequency are consistent and significant. Both RCP2.6 and RCP8.5 show more frequent compound drought events (see Fig. 6). Projected changes in frequency, significance, and model agreement are smaller and less consistent under RCP2.6 (Fig. 6a–d). By 2085, a compound drought event is projected to occur in (multi-model) median every 1.8 to 2 years north of the Alps while events remain rarer in southern Switzerland with an event occurring slightly more often than once in four years (every 3.9 years) (Fig. 6d). This corresponds to an overall (multi-model) relative median increase of less than 50% (Swiss Plateau: +42.2%; Jura: +28.6%; Southern Switzerland: +12%) except for the Pre-Alps (+68.4%) (Fig. 6k). Model agreement is higher for the Pre-Alps, southern Switzerland and the Swiss Plateau region with at least 75% models agreeing on more frequent events whereas changes are more uncertain in the Jura region (only 62.5% models show more frequent events).

Under RCP8.5, projected increases in frequency become significant by 2035 for both Jura and Pre-Alps with almost general model agreement in the Pre-Alps (85% of models agree) (Fig. 6f). By 2035, compound drought events are projected to occur in (multi-model) median every 1.6 to 1.8 years in the Jura region and the Pre-Alps, every second year in the Swiss Plateau region and slightly more often than every four years (every 3.7 years) in southern Switzerland. This corresponds to 58% more frequent compound drought events in the Pre-Alps by 2035 (Fig. 6m). By mid-century, significantly more frequent compound events are projected in all Greater regions and models do generally agree in all regions but the Swiss Plateau region (Fig. 6g). Compound drought events are projected to occur every 1.25 years in the Jura region, every 1.3 to 1.4 years in both Swiss Plateau and Pre-Alps and every 2.7 years in southern Switzerland. This corresponds to an almost three-fold (+178.7%) increase in compound drought events in the Pre-Alps and more than a doubling in the Swiss Plateau region (+113.3%) in (multi-model) median by 2060 (Fig. 6m). Percental increases are lower for southern Switzerland (+71%) and the Jura region (+62.9%) where models show less than a doubling in frequency. With the more subtantial and significant increase, also the multi-model distributions between mitigation and non-mitigation scenarios begin to differ more clearly at mid-century and become significantly different by the end of the century for all Greater regions (Fig. 6h). A compound drought event is projected to occur at least once per extended summer season in regions north of the Alps with highest frequencies in the Pre-Alps (median 1.43 events / every 0.7 years) and slightly lower frequencies in the Swiss Plateau (1.28 events / 0.77 years ) and Jura (1.13 events / 0.88 years ) regions (Fig. 6h). In catchments in southern Switzerland, compound events are projected to occur in median in almost three out of four




## Compound drought event duration

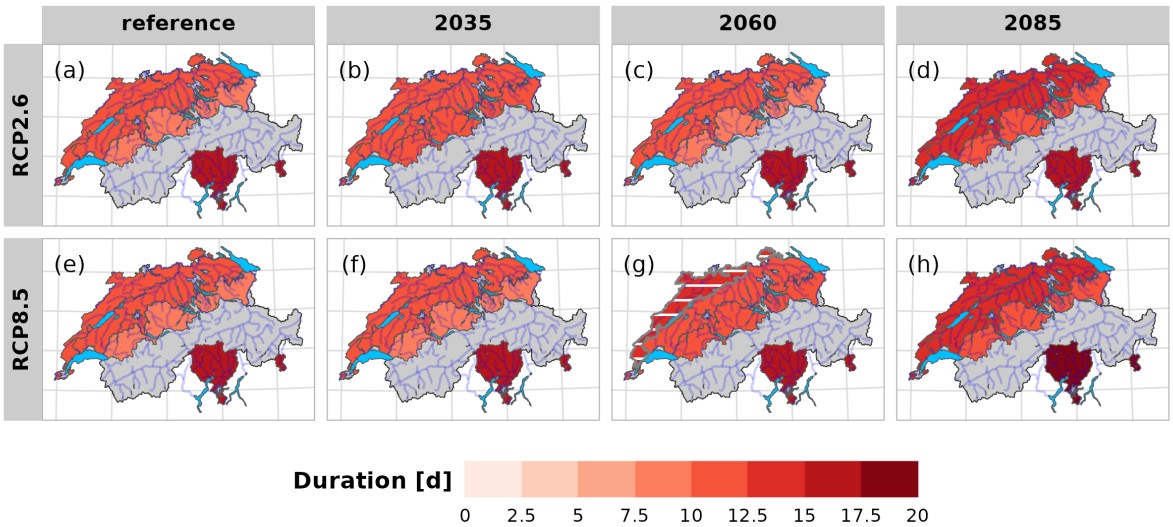

## Percentage change

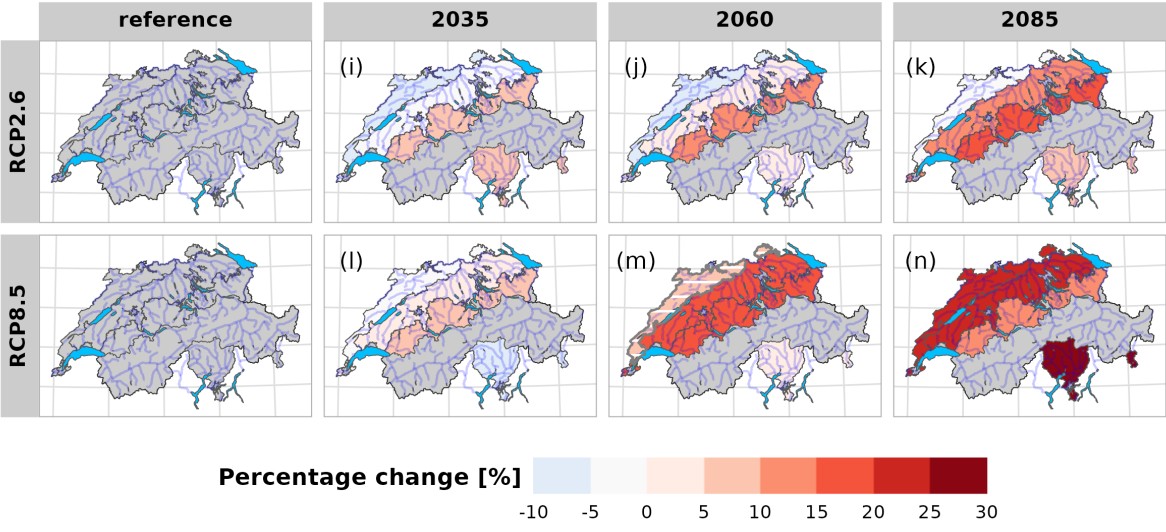

**Figure 5.** Compound drought event duration per extended summer season and projected changes under climate change for a mitigation (RCP2.6) and non-mitigation (RCP8.5) scenario aggregated across the Greater regions (median over all catchments). Presented are both median value (top) and percentage change (bottom) over all model simulations per scenario. Hatching indicates if changes are significant compared to the reference period (horizontal lines) and whether ≥ 90% of model simulations agree on the projected changes (per scenario; vertical lines). Significance and model agreement were evaluated only on absolute values and are identical for both median values and percentage changes.





years (every 1.36 years / 0.73 events) which is less compared to north of the Alps but still represents a major increase compared
to the reference period. Projected multi-model percental increases are strongest in the Pre-Alps with around four-and-a-half
times (+352.7%) more frequent events, followed by three-and-a-half times more frequent events in the Swiss Plateau region
(+248%), approximately three-times more frequent events in southern Switzerland (+188%) (Fig. 6n). In the Jura region, the
increase corresponds to two-and-a-half times more frequent events (+153%).

### 4.3.1  Seasonality

In the reference period, the seasonality of compound drought days is similar among all regions north of the Alps with peak
occurrence probabilities between mid-July and mid-September (Fig. 7a–c,e–g). Under RCP8.5, the probability density dis-
tributions are flatter, and higher probabilities for compound drought day occurrence extend to the begin of October (Fig.
7e–g). Compound drought events are more evenly distributed across the summer months in the Jura region compared to the
other regions, especially under RCP2.6. By 2085 and without mitigation, the probability of compound drought day occurrence
increases mainly between mid-August to mid-September, but remains generally high in the entire period from July to mid-
October (see Fig. 7e–g). In contrast, a decrease in probability density is projected starting towards the end of the extended
summer season in fall extending through winter to mid-March. Overall, projected changes under RCP8.5 lead to a more pro-
nounced seasonality of compound drought days with higher occurrence probabilities in the critical summer months and early
autumn (July–September, see e.g., Fuhrer and Jasper, 2009). The analysis of absolute frequencies of compound drought days
indicates that the shift in seasonality is caused by a combination of decreases outside and increases within the extended summer
season (not shown). Projected changes under RCP2.6 show similarities with changes under RCP8.5 (e.g. decrease outside of
the extended summer season) but are generally not sufficiently consistent or evident (Fig. 7a–c).

In southern Switzerland the seasonality of compound drought days is distinctly different from the regions north of the Alps
(Fig. 7d,h) following the bi-modal seasonality of hydrological droughts with periods of higher probability both in summer
and winter(not shown; see e.g., Muelchi et al., 2021b). A first period with high probability of compound drought occurrence
ranges from the begin of January to mid-April with the probability peaking around mid-February. A second period of higher
probabilities of occurrence starts approximately in mid-July and extends towards the end of the year. Overall, more than 50%
of compound drought days occur outside of the extended summer season. Compared to regions north of the Alps, compound
drought days in southern Switzerland are generally more equally distributed across the year and the peak probabilities in the
extended summer season is consequently less pronounced (Fig. 7d,h). Under RCP8.5, the probability of compound drought
day occurrence strongly increases between the begin of July and end of September while strong decreases are projected outside
of the extended summer season most prominently between January and mid-April (Fig. 7h). Both decreases outside and strong
increases within the extended summer season will lead to a gradual reversal of the bi-modal seasonality distribution towards
the end of the century (Fig. 7h). For southern Switzerland, the absolute compound drought day frequencies indicate that the
changes are more strongly driven by increases in frequency during the extended summer season and decreases in absolute
frequencies outside of the extended summer season first become evident by the end of the century (not shown). Without



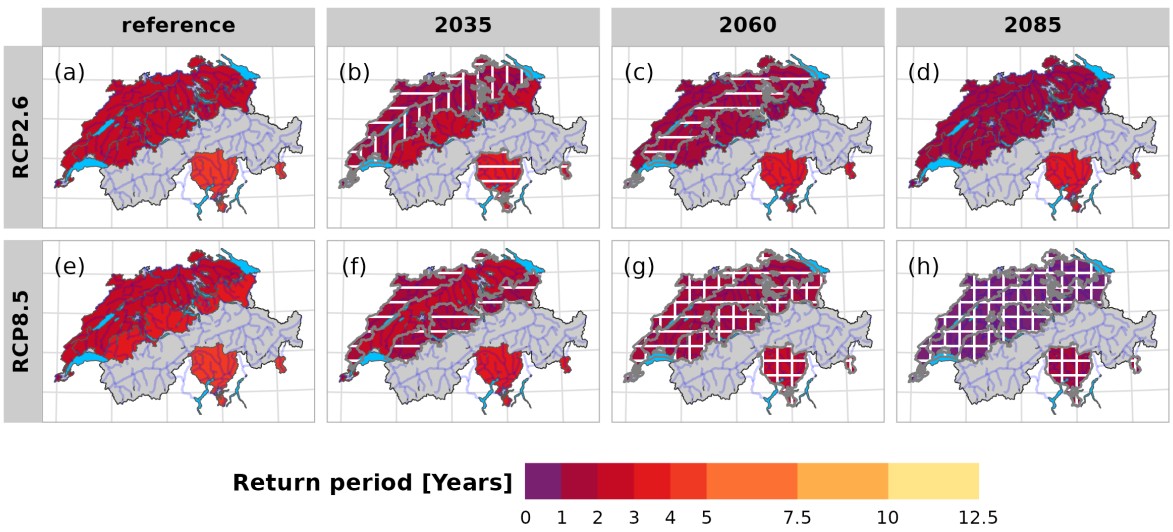

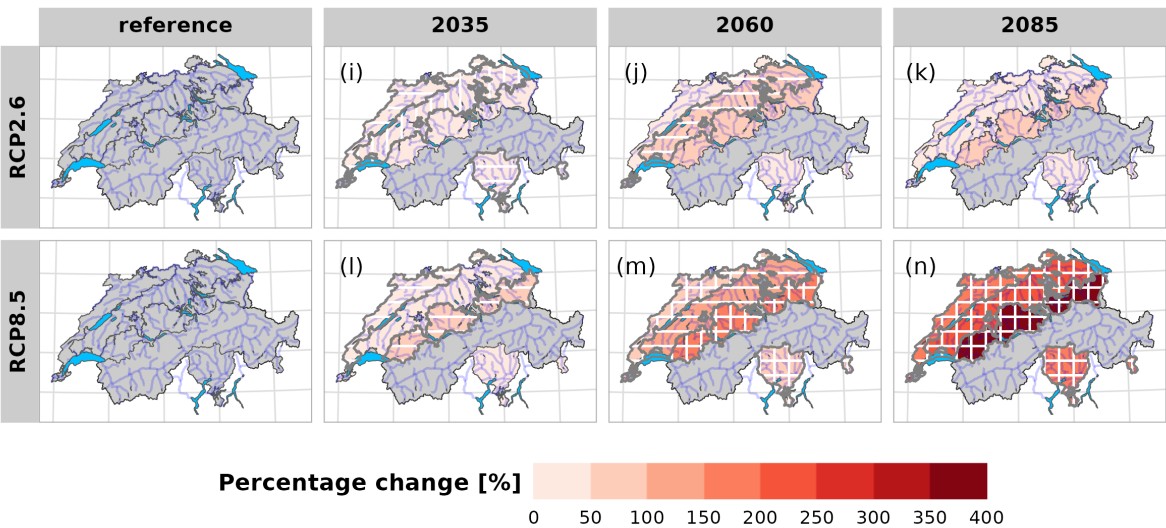

**Figure 6.** Compound drought event return periods and projected changes under climate change for a mitigation (RCP2.6) and non-mitigation (RCP8.5) scenario aggregated on the Greater regions (median over all catchments). Presented are both median value (top) and percentage change (bottom) over all model simulations per scenario. Hatching indicates if changes are significant compared to the reference period (horizontal lines) and whether ≥ 90% of model simulations agree on the projected changes (per scenario; vertical lines). Significance and model agreement were evaluated only on absolute values only and are identical for both median values and percentage changes.




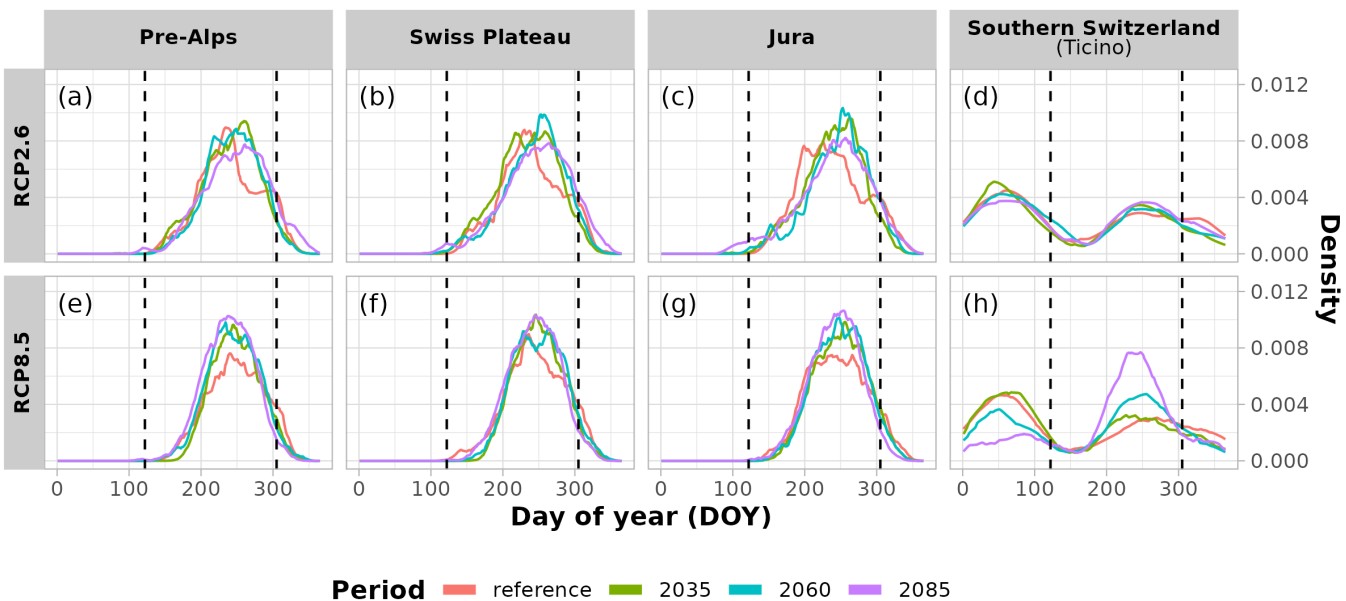

**Figure 7.** Seasonality of compound drought days: shown is the median value aggregated over all catchments per Greater regions and day of year (DOY) (lines). Shading indicates the IQR-range of all model chains. Density distributions are shown for both the mitigation (RCP2.6) and the non-mitigation (RCP8.5) scenario for the periods reference, 2035, 2060, 2085 (colored). Dashed lines indicate the extended summer season from (begin of) May (DOY 122) to (the end of) October (DOY 305).

.

mitigation, compound drought day occurrence is projected to become most probable between begin to mid-July until begin of October and by this shift becomes more aligned with seasonality of regions north of the Alps. Again, similarities exist under
RCP2.6 but remain less evident (Fig. 7d).

## 5    Spatial extent of multivariate compound droughts (*spatially compounding droughts*)

### 5.1    Median number of simultaneously affected catchments

In the reference period, more than 10 catchments simultaneously affected in about 20% (RCP8.5) to 24% (RCP2.6) of all compound drought days (Fig. 8a). The number of simultaneously affected catchments per compound drought day is projected
to increase in the future. The increase is more pronounced in the non-mitigation scenario RCP8.5 (Fig. 8a). By 2085 and without mitigation, in median on 46% of days at least 10 catchments are simultaneously affected by compound drought conditions, on 27% of days at least 20 catchments, and in median on 5.8% of days more than 40 catchments (Fig. 8a). The projected increase in the median number of catchments simultaneously affected by compound drought conditions is also reflected in the





distributions over all drought days (see Fig. 8b). In the reference period, 4-5 catchments are simultaneously under compound

drought conditions. Under RCP8.5 the number of simultaneously affected catchments in near future (2035) increases to 6 and

the increase is significant. However, only 55% (11/20) of models agree on an increase. Model agreement gradually increases

and reaches general model agreement (95%) by the end of the century (Fig. 8b). By 2085, in median 8.5 catchments are

projected to be simultaneously under compound drought conditions on a compound drought day. Note that the possible range

of projected increases is large (min-max: 6–25 catchments; Fig. 8b). Comparing both mitigation and non-mitigation scenarios,

multi-model distributions are however not significantly different for all periods (Fig. 8b).

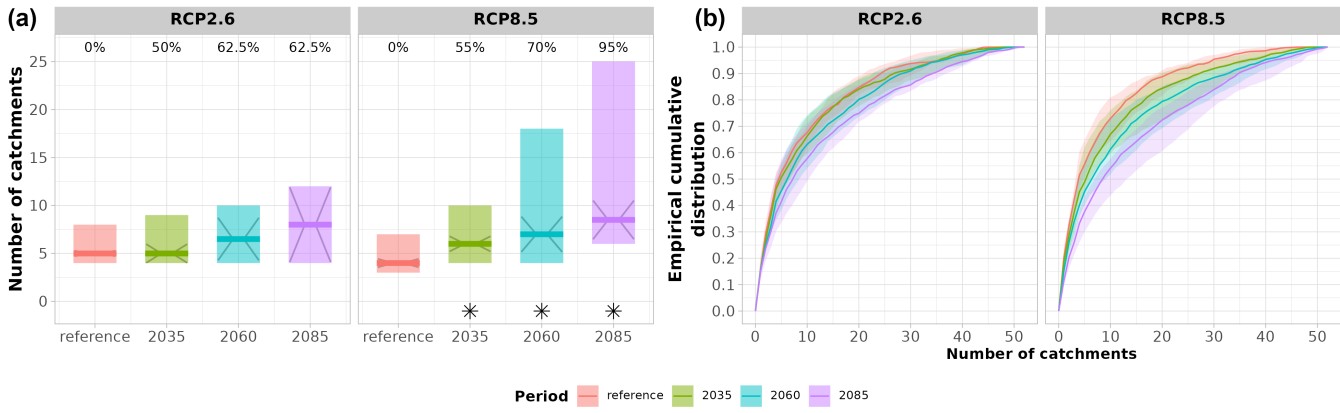

**Figure 8.** Spatially compounding droughts and catchments simultaneously affected by compound drought conditions. **(a)** Median number of catchments simultaneously affected by compound drought conditions on a specific compound drought day and projected changes under climate change. Color shading indicates the full range of (median) realisations (min-max). The median over all model simulations is indicated by thick color lines. Confidence intervals (CIs, see section 3.3) of the median are shown with diagonal lines originating from the center of the median. Significance based on strict non-overlapping of CIs is indicated by asterisks. Model agreement is shown in percentage values on top. **(b)** Cumulative distribution of the number of catchments simultaneously affected catchments by compound drought conditions on a specific compound drought day. Color shading shows the IQR of cumulative distributions over all model simulations per scenario. The data shows distributions for both mitigation (RCP2.6) and non-mitigation (RCP8.5) scenario and periods 2035, 2060 and 2085.

## 5.2   Event duration and maximum number of simultaneously affected catchments

As for spatially compounding days, most events affect in maximum 10 or fewer catchments (see Fig. 9). The median number of spatially compounding events is comparable among scenarios and does not differ by more than 10% in most cases. Exceptions are shorter events affecting less catchments where 22.6% more events are simulated under RCP8.5 and events affecting be-

tween 30–40 catchments simultaneously (+42.9% more events under RCP8.5). Events potentially affecting all Greater regions simultaneously occur in median only 2 times per model in the entire time series 1991–2100 under RCP2.6 and approximately two-and-a-half times more (5 times) under RCP8.5 (Fig. 9).

Spatially compounding events affecting at maximum 10 catchments simultaneously last in median 9 days in the scenario-





driven simulations and with approximately every two weeks (12–14 days) of persistence, 10 additional catchments transition

into compound drought conditions (Fig. 9). Under RCP8.5, events tend to be longer in median. The distributions do however not significantly differ in median from RCP2.6 for events affecting in maximum more than 20-40 catchments. Differences in median exist for events affecting at least 3 Greater regions (40–48 catchments) simultaneously with events lasting in median 56 days under RCP2.6 and 66 days under RCP8.5, respectively. More pronounced are differences for events affecting all Greater regions and thereby potentially affecting entire Switzerland (48–52 catchments). Potentially countrywide events may persist

in median 65.5 days under RCP2.6 and almost 40% longer under RCP8.5 with a median event duration of 91 days (Fig. 9). Hence, there is an indication that spatially compounding drought events may potentially last longer without mitigation or that more and potentially longer countrywide events may occur in future.

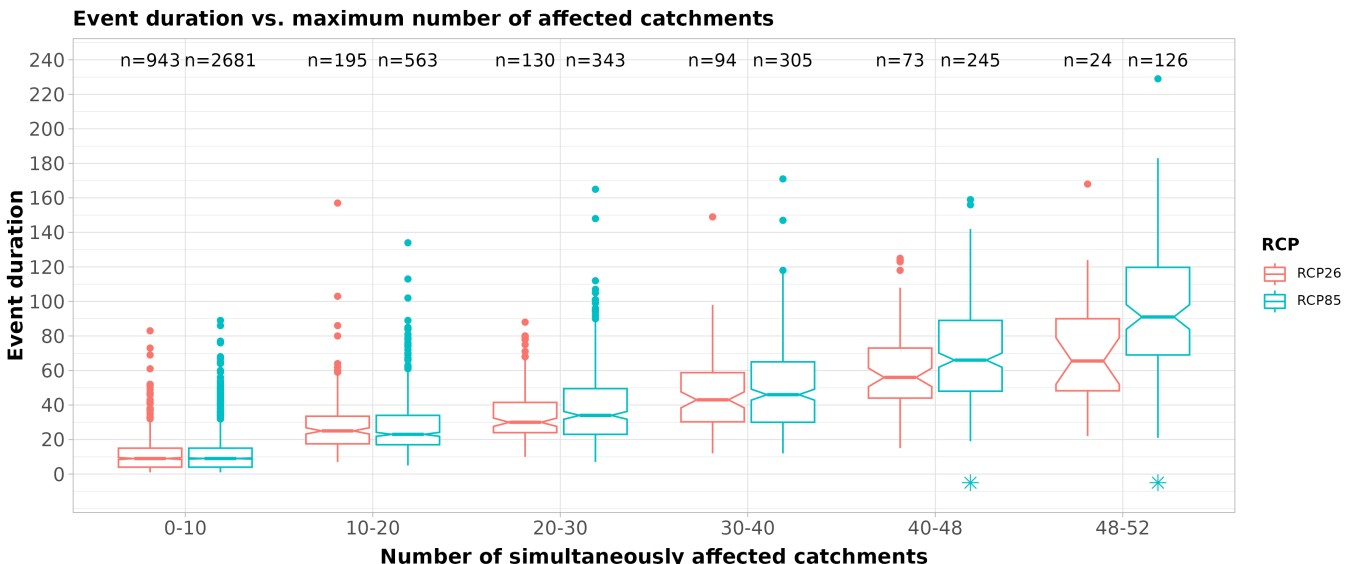

**Figure 9.** Spatially compounding drought events, event duration and maximum number of simultaneously catchments affected by compound drought conditions. The boxplots show the distributions of event durations conditional on the maximum number of simultaneously affected catchments for both the mitigation (RCP2.6) and the non-mitigation (RCP8.5) scenario. Significant differences are indicated with asterisks. Significance was assessed by comparing confidence intervals of the median (*notches*, see section 3.3) of RCP8.5 median distributions with CIs of RCP2.6 distributions for each range of simultaneously affected catchments. Number of events contributing to distributions is indicated by "n=".

## 6  Discussion

Changes in individual drought types have been found for meteorological droughts (CH2018, 2018; Kotlarski et al., 2023;

Scherrer et al., 2022), agricultural droughts (CH2018, 2018; Remund et al., 2016; Tschurr et al., 2020; Calanca, 2007) and hydrological droughts (Brunner et al., 2019b; Brunner and Tallaksen, 2019; Muelchi et al., 2021b; Brunner et al., 2023). This





study shows that also the concurrent occurrence of all three drought types is projected to increase significantly under climate change in all regions of Switzerland. The simultaneous deficit of water in the atmosphere, the soils, and in the rivers may aggravate impacts of droughts. Compound drought characteristics resemble characteristics of the hydrological drought which
occur least frequently among individual drought types and may be a limiting factor for compound drought events. This is most prominent for seasonality, the number of compound drought days and event frequency. The relative differences among Greater regions however weaken by the end of the century as the seasonality aligns across all Greater regions and more days and more frequent events occur in absolute terms. A difference to hydrological droughts exists for event duration with longer compound drought event durations in catchments in north-western Switzerland and especially in southern Switzerland. In
southern Switzerland, compound drought events occur 40-50% less frequently, but last 45-77% longer compared to northern Switzerland. The longer duration of compound droughts may be a result of higher event frequencies and event duration in agricultural drought in both southern Switzerland and the Jura region.

The differences among Greater regions are generally consistent with other studies showing that the Jura and Swiss Plateau region and southern Switzerland are more strongly affected by projected changes in drought conditions (Brunner et al., 2019a;
Hirschi et al., 2020; Kohn et al., 2019; Tschurr et al., 2020; Walthert et al., 2015). Further, catchments with largest numbers of compound drought days or event frequencies are also found to be prone to water scarcity in other studies (Brunner et al., 2019a; Fuhrer and Jasper, 2012; Fuhrer and Calanca, 2014; Lanz, 2020). Given that in extreme summers (e.g., 2003, 2018), many catchments do already face water scarcity (Brunner et al., 2019a) and that our results are in line with with other studies, we hence consider our results plausible. This holds especially true as extreme summers at present are projected to become
more average with no mitigation in future (Imfeld et al., 2022b, a; Calanca, 2007).

Our findings regarding the more widespread occurrence of (spatially compounding) compound droughts in future is likely related to the general alignment of seasonality in compound droughts in Switzerland (see Section ??). Compound drought seasonality appears strongly linked to streamflow regimes which further underlines that aggravation of water scarcity or resulting challenges in water management are also related to streamflow regimes (Brunner et al., 2019a; Fuhrer and Calanca, 2014).
Streamflow transitioning towards more lowland characteristics is projected in the Pre-Alps as reaction to earlier snow melt and less snow melt contribution (Brunner and Tallaksen, 2019; Muelchi et al., 2021b). Although we restricted our analysis to lower lying catchments with limited snow influence, this shift is visible especially for pre-alpine catchments closer to the main Alpine ridge covering large elevation ranges. For the other regions a pronounced seasonality comprising the critical summer months and early autumn (July-–September) is visible by 2085 under RCP8.5 and is consistent with projected changes in streamflow
regimes and characteristics of individual droughts (see e.g., BAFU (Hrsg.), 2021; CH2018, 2018; Hirschi et al., 2020; Remund et al., 2016; Walthert et al., 2015; Brunner et al., 2019b; Muelchi et al., 2021b). Further, also the relationship between the duration of spatially compounding drought events and the number of simultaneously affected catchments is consistent with the principles of spatial drought propagation (spatial spreading of droughts): the longer an event persists, the more catchments gradually experience compound drought conditions simultaneously due to catchment-specific response times to the drought
signal (Fig. 9;Van Loon and Van Lanen (see e.g., 2012); Apurv and Cai (see e.g., 2020); Tijdeman et al. (see e.g., 2022);





Van Loon et al. (see e.g., 2014); Ho et al. (see e.g., 2021); Zhang et al. (see e.g., 2022); Sutanto and Van Lanen (see e.g., 2022); Van Loon and Laaha (see e.g., 2015)).

Drought projections based on climate-hydrological model chains contain major sources of uncertainty, which should be considered when interpreting the results of this study. This includes choices of model ensembles, (hydrological) model-setup,
the representation of land-atmosphere interactions (e.g., plant-physiological processes), choice of drought index, multi-decadal variability, drought propagation, and catchment storage properties (Arias et al., 2021; Berg and Sheffield, 2018; Brunner et al., 2021; Lehner et al., 2017; Miralles et al., 2019; Orlowsky and Seneviratne, 2013; Scherrer et al., 2022; Vicente-Serrano et al., 2022).

We use standard indices for both meteorological and agricultural drought, and a widely used streamflow index closely related
to water regulations in Switzerland. However, establishing direct links between drought indices and drought impacts is still challenging due to the scarcity of impact data and the extended duration of drought events, which make associations of impacts more challenging (Bachmair et al., 2016; Kchouk et al., 2022; Merz et al., 2020). Recent research on impact-relevant soil moisture thresholds and cumulative water deficits may lead to more consistent agricultural drought definitions in future (Denissen et al., 2020; Fu et al., 2022b; Stocker et al., 2023).

Implementations of evaporative responses vary among hydrological models and can have influence on climate change projections (Melsen and Guse, 2019). Recent findings indicate that (agricultural) drought projections that use temperature-only based estimations of PET result in the overestimation of future drying (Berg and Sheffield, 2018; Milly and Dunne, 2017, 2016). Comparisons of our simulations with hydrological simulations based on Penman-Monteith parameterisation conducted with the same model (PREVAH) did however not show significant differences (R. Muelchi, pers. msg.) and projected changes do
not deviate substantially (BAFU (Hrsg.), 2021). Markonis et al. (2021) further highlight that (agricultural) drought conditions are rather driven by actual evapotranspiration than solely by PET and is thus dependent on actual (soil) water availability and plant evaporative responses. By incorporating actual ET, a measure of actual water availability is included by using ET/PET (Fuhrer and Jasper, 2009). The validation of soil moisture and evapotranspiration is however often challenging due to limited availability of long-term observational data (Haile et al., 2020; Mukherjee et al., 2018; Hirschi et al., 2020). Agricultural
droughts relying on soil moisture or evaporation data derived from hydrological model simulations driven by meteorological forcing data must thus generally be interpreted with caution (Mukherjee et al., 2018). We however emphasize that PREVAH has shown to be able to represent the water balance of Switzerland, was used in similar water scarcity studies (Brunner et al., 2019a, and sources therein) and has further shown to perform well also in extreme situations outside of observational ranges (Zappa and Kan, 2007).


To validate of our results for the investigated multivariate compound drought characteristics, the hydrological control simulation driven by observations (CTRL) was used and compared with the multi-model median ranges (see *Supplementary Material*). Characteristics derived from CTRL were mostly within the range of our model ensemble and if not, deviations were generally small with values in the same order of magnitude. CTRL-run characteristics were generally better in line with
RCP8.5 multi-model distributions. Direct comparisons with observations and an information on uncertainty of both Hydro-





CH2018 and CH2018 datasets regarding SPI and streamflow can be found in Tschurr et al. (2020) and Muelchi et al. (2022). For agricultural drought, comparisons with studies using a similar approach with either the same hydrological model and a different measure for water demand or vice versa revealed comparatively good agreement in both magnitude orders of drought characteristics (see e.g., Fuhrer and Jasper, 2009; Walthert et al., 2015) and spatial patterns (see Brunner et al., 2019a; Fuhrer and Jasper, 2012, 2009; Hirschi et al., 2020; Remund et al., 2016; Walthert et al., 2015) given the associated uncertainties and in most cases limited comparability of our catchment-scale data to field-scale or point-location data. Also note that Fuhrer and Jasper (2009) showed that for agricultural land, the average ET/PET values are close to our and their threshold of ET/PET 0.8 already at present. Here, we analysed ET/PET on catchment-level which does not explicitly target a specific land use. Already at present much agricultural land in Switzerland falls regularly below the critical threshold for several weeks or even months and average ET/PET values are projected to be considerably lower under a non-mitigation scenario in future (Allgaier Leuch et al., 2017; Remund et al., 2016; Walthert et al., 2015). An analysis targeting specific land use types (e.g. agricultural land) could consequently lead to more severe compound drought projections than reported here. Also, we do not account for distance to surface waters which can make irrigation by surface water abstraction unprofitable.

Van Loon (2015) suggest that the combined occurrence of agricultural and hydrological droughts without meteorological drought conditions is sufficient for impacts on irrigated agriculture. We also conducted a brief analysis on this type of compound drought event. Our analysis shows that combined agricultural and hydrological drought characteristics are qualitatively similar to the results discussed here but the absolute number of compound drought days and event frequency is higher and events last longer (*see Supplementary Material*).

The main goal of this study was to assess characteristics of compound droughts and changes under climate change. We therefore mainly grouped catchments by similarity of streamflow regime types and did not explicitly consider spatial proximity or catchment connectivity. Future studies on compound drought events and impacts on water management may focus on specific source regions relevant for water management actions. Further, hydrological catchments have been calibrated independent of each other but can in some cases be part of a larger catchment (Muelchi et al., 2022). Future studies could thus also investigate the temporal evolution of spatially compounding droughts and their downstream propagation behaviour by hydrological simulations of coupled (sub-)watersheds.

A limitation of the present hydrological model simulations of the Hydro-CH2018 runoff ensemble is, that they assume stationarity in many influence factors such as land cover, land use practices and catchment storage characteristics (Savelli et al., 2022; Brunner et al., 2021; Melsen and Guse, 2019). Changes in drought propagation characteristics (e.g., onset and recovery) are likely to occur with climate change due to (projected) changes in catchment properties, drought generating processes (Mukherjee et al., 2018; Zhang et al., 2022; Brunner and Tallaksen, 2019; Brunner et al., 2023) and climate characteristics (e.g., precipitation frequency and intensity) (CH2018, 2018; Kotlarski et al., 2023; Tschurr et al., 2020; Eekhout et al., 2018). Implementations of drought triggering processes vary among hydrological models and important storage variables are often parameterized (Melsen and Guse, 2019; Brunner et al., 2021). More research on the representation and a better understand-





ing of key processes, storages, human interactions, and their influence on characteristics of drought events and propagation characteristics in hydrological models is therefore crucial for future climate change assessments which optimally rely on multi-hydrological climate ensembles (see e.g., Melsen and Guse, 2019), account for non-stationarity (see e.g., Brunner et al., 2021) and possibly make use of climate scenarios of the Coupled Model Intercomparison Project Phase 6 (CMIP6, Eyring et al., 2016) and Shared Socioeconomic Pathways (SSPs, O'Neill et al., 2014).


## 7 Conclusions

Compound droughts defined here as concurrent meteorological, agricultural and hydrological droughts pose a challenge to the water management in Switzerland because the water abstraction from rivers is limited during low flow periods and while farmers need water for irrigation during agricultural drought conditions. We analyse projected changes in frequency, duration and

spatial extent of compound droughts using data from numerical climate model simulations CH2018 (2018) and from hydrological model simulations Muelchi et al. (2022) for 52 catchments in Switzerland. We compare ensemble projections based an emission scenario with mitigation (RCP2.6, 8 model chains) and an emission scenario without mitigation (RCP8.5, 20 model chains). The key findings are:

– The number of days with compound meteorological, hydrological, and agricultural droughts in Switzerland are projected to increase significantly by the end of the $21^{st}$ century in an emission scenario without mitigation in all greater regions of Switzerland. There is a broad agreement among the model simulations on the sign of the change. The number of days with compound meteorological, hydrological, and agricultural droughts in Switzerland does not increase significantly by the end of the $21^{st}$ century in an emission scenario with mitigation. With mitigation the number of compound drought

days is reduced by 50–55% north of the Alps and by up to 75% in the Southern Alps pointing to the importance of mitigation.

– The number of days with compound droughts increases predominantly during the extended summer and hence the main agricultural production season.

– The increase in the number of compound drought days is driven by an increasing number of events (significant increase)

rather than by longer lasting events (no significant change). Across all model chains compound drought events are projected to occur on average once per summer in the catchments on the Alpine northside and once every 1–2 summer seasons in the catchments on the Alpine southside.

– Coupled to the projected increasing number of compound droughts days we find that significantly more catchments are affected at the same time.

The results hence point i) to the benefits of mitigation measures taken at an early stage as distributions among scenarios begin to differ by mid-century (2060) and ii) to the need for coordinate adaptation as drought days that might affect agricultural



production due to lack of soil moisture combined with potential water abstraction limitations for irrigation are projected to occur on average once a summer. This strongly emphasizes the importance of mitigation measures, especially considering recent findings that adaptation alone is not always sufficient to avoid severe impacts (Kreibich et al., 2022; Tellman and Eakin, 560 2022).

*Data availability.* The dataset is available from Zenodo (https://doi.org/10.5281/zenodo.10377585). For the original runoff simulations see Muelchi et al. (2022). For the CH2018 climate scenarios see CH2018 Project Team (2018). Data for the Greater regions was kindly provided by the FOEN and can be requested via climate-adaptation@bafu.admin.ch . Map data is provided by Swisstopo and freely available from https://www.swisstopo.admin.ch/de/geodata/height/dhm25.html .

*Author contributions.* CM conceptualized and performed the formal analysis and drafted the article. OM, RM and LG provided guidance on the methodological aspects and treatment of projection uncertainty. All authors assisted with paper writing and revisions. RM conceptualized the project proposal and acquired funding from FOEN.

*Competing interests.* The authors declare that they have no conflict of interest.

*Acknowledgements.* This research was supported by the Swiss Federal Office for the Environment (FOEN) under the project COM-DROUGHTS. 570 CM and ORM acknowledge support from the Mobiliar Lab for Natural Risks. The authors further thank Vincent Roth (FOEN) for providing the Greater regions data used for aggregation. Data for the overview map of the study region (Fig. 1) is freely available from the Federal Office of Topography (Swisstopo) (digital elevation model). For spatial visualization the *sf*-package (Pebesma, 2018) was used.



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
