# Peer review of "Compound droughts under climate change in Switzerland"

_Natural Hazards and Earth System Sciences, 2024_

## Author Comment (AC1)

**Authors Response to peer-reviews for nhess-2024-6**

Christoph von Matt, Regula Mülchi, Lukas Gudmundsson, Olivia Martius

First we want to thank the two anonymous reviewers for taking their time to review our manuscript on compound droughts under climate change in Switzerland. Their feedback was very encouraging and helpful in terms of a more complete contextualization of the results and for (editorial) improvements and redactions.

The answers to the suggested improvements are colored in *Android Green*, new/altered parts are colored in *Azure*. Original line numbers and original reviewer comments/suggestions are colored in *Black*, while line numbers in *Deep carmine* [and in brackets] denote new line numbers to allow for change tracking Deleted parts are colored in *red* and are crossed-out (e.g., ).

**Review 1**

**Comment on nhess-2024-6 (Anonymous Referee #1)**

Referee comment on "Compound droughts under climate change in Switzerland" (nhess-2024-6) by von Matt et al., Nat. Hazards Earth Syst. Sci. Discuss., https://doi.org/10.5194/nhess-2024-6-RC1

Review of "Compound droughts under climate change in Switzerland" von Matt et al.

The topic of compound droughts is pressing and important, and the authors have contributed very nicely in identifying them and using scenarios to see how they might change in a warming climate, using transient climate and hydrological scenarios for Switzerland.

The paper is clearly structured and well written and provides a comprehensive and convincing introduction to the topic.

Thank you very much.

For some of the underlying assumptions and control model performance tests, the authors refer to previous studies (catchments and calibration validation performance). I think it is important to provide this information directly in the paper or in its appendix to understand the basis on which the scenarios were used and how reliable the models were. Looking at the referenced paper, it appears that the NSE was used

to calibrate the models and the KGE and NSE were used to validate them. I would expect that particularly low flow periods would not be so well simulated, or at least not evaluated in the objective function. Perhaps this could also be mentioned in the discussion of uncertainty in the discussion.

As a minimum I would expect a table with the main characteristics of the catchments used, rather than just a map, including mean annual precipitation, mean annual discharge, altitude (range), glaciation percentage, size and some model performance in calibration and validation. The authors state at the end of their discussion that it is important to better understand the processes that lead to these compound droughts, and I believe that to understand these we also need to look at these catchment characteristics and therefore provide them in some form as well.

Thank you very much for this suggestion. We completely agree with you and have made the following adjustments:

1. A table (see Table 1) with catchment and climate characteristics is added to Appendix A of the manuscript and a separate table (see Table 2) with calibration/validation characteristics is added to the Appendix B of the manuscript. The table consists of all requested characteristics. A corresponding note on glaciation was added in the section *2 Data* as well. For consistency, we derived annual precipitation and annual runoff based on the observation-driven runoff simulations similar as was done for the validation of our compound drought characteristics (see Section *2 Data* and *Supplementary Material*) based on the longest overlapping period (1991–2014) within the reference period (1991–2020). Therefore, the mean annual precipitation is directly derived from catchment-level aggregated values originating from the observational MeteoSwiss RhiresD-dataset, while mean annual runoff is derived from the observation-driven PREVAH-simulations. A short description on presented metrics and climate characteristics are also provided as footnote to the corresponding tables. References to the table in the Appendix A and B were added on:

   Lines 126–128 [132–136]: "The restriction to lower lying small- to mid-size catchments and to the extended summer season ensures a focus on classical rainfall deficit droughts and limited influence of snow- and glacier-melt on streamflow discharge (Brunner et al., 2019b; Floriancic et al., 2020; Muelchi et al., 2021a, b). As such, all investigated catchments have a glaciation percentage of 0 %. A more detailed description of the investigated catchments including catchment and climate characteristics is presented in Table A1 in Appendix A."

2. Further, we included a more detailed description of the calibration and validation procedure, directly mentioning the key terms incorporated in the weighting scheme. The following sentences were added:

Line 139 [149–163] : " The hydrological PREVAH-model was calibrated following the automated parameter estimation procedure PEST (Doherty, 2005) by using the objective function ($\Phi$) which is defined as the squared sum of weighted residuals (see Muelchi et al., 2022):

$$\Phi = \sum (w_i \times r_i)^2 \qquad (1)$$

where $r_i$ denotes the residual of the $i^{th}$ observation and $w_i$ the weight associated with the $i^{th}$ observation.

Four equally weighted observation groups were considered: 1) the observed runoff ($Q$), 2) the monthly mean runoff ($Q_{month}$), 3) the yearly volumes ($Q_{year}$) and 4) a transformed (inverted) runoff ($(max(Q) + min(Q)) - Q_i$) to add more weight to low flow conditions. Therefore, the objective function is conditioned towards a better representation of river flow regimes and low flow conditions (R. Muelchi, pers. com.; see also Muelchi et al., 2022). Performance was assessed by both the Nash-Sutcliffe (NSE, Nash and Sutcliffe, 1970) and Kling-Gupta (KGE, Gupta et al., 2009) Efficiency for calibration and validation periods separately. Calibration and validation metrics for all investigated catchments are presented in Table B1 in Appendix B. For more details see Muelchi et al. (2022)."

3. Lastly, we also complemented the discussion with a paragraph on model calibration/validation uncertainty and challenges of current hydrological models concerning the representation of low flows with emphasis on influences of fixed-storage assumptions of evapotranspiration and corresponding interactions with low flows / hydrological droughts realism in (present and) future climates.

The following additions were made with regard to model calibration/validation:

Lines 458–463 [480–490]: "Drought projections based on climate-hydrological model chains contain major sources of uncertainty, which should be considered when interpreting the results of this study. This includes choices of model ensembles, (hydrological) model-setup (including calibration and validation procedures), the representation of land-atmosphere interactions (e.g., plant-physiological processes), choice of drought index, multi-decadal variability, drought propagation, and catchment storage properties (Arias et al., 2021; Berg and Sheffield,

2018; Brunner et al., 2021; Lehner et al., 2017; Miralles et al., 2019; Orlowsky and Seneviratne, 2013; Scherrer et al., 2022; Vicente-Serrano et al., 2022).

Current hydrological model calibration and validation mostly rely on calibration and validation metrics designed for specific applications (e.g. flood situations, Brunner et al., 2021). The Hydro-CH2018 hydrological scenarios use a multi-objective calibration scheme also accounting for the representation of low flow conditions (see Section 2; see also Muelchi et al., 2022). See Table B1 in Appendix B for validation metrics ($NSE_{log}$ and $KGE_{log}$) indicative of the representation of low flow conditions for all catchments."

Lines 513–516 [540–544]: "Further, hydrological catchments have been calibrated independent of each other but can in some cases be part of a larger catchment (Muelchi et al., 2022). Future studies could thus also investigate the temporal evolution of spatially compounding droughts and their downstream propagation behaviour by hydrological simulations of coupled (sub-)watersheds and/or by accounting for spatial connectivity by incorporating a spatial calibaration/validation metric (Brunner et al., 2021)."

The following additions related to the influence of the representation of evapotranspiration on low flow modelling were added:

Lines 524–525 [552–559]: "Implementations of drought triggering processes vary among hydrological models and important storage variables are often parameterized (Melsen and Guse, 2019; Brunner et al., 2021). Assumptions on (fixed) (maximum) storage volumes in hydrological models are equivalent to an implicit limitation on deficit accumulation. Redesigned soil moisture storage implementations could therefore lead to more realistic hydrological model projections in future (Fowler et al., 2021). Improved realism in projections is of utmost importance with regard to recent studies highlighting the potential for shifts in catchment-specific rainfall-runoff relationships usually caused by (prolonged) multi-year droughts (Fowler et al., 2022; Brunner and Tallaksen, 2019; Saft et al.,2015). Consequentially, climate risk assessments based on hydrological model projections might underestimate the future hydro-climatic risk concerning reductions in water supply (Fowler et al., 2022)."

Other than that, I really enjoyed reading the paper and have very few and small technical comments:

Thank you very much for this kind/honoring comment!

L135 please add what the Hamon equations use as main variables to calculate PET (lon, lat, air temperature)

We added the following information:

Lines 137–139 [145–149]: "PET is calculated by the Hamon equations, which is a temperature-based estimation method which derives average PET based on the saturated water vapor density at the daily mean temperature adjusted for the number of daylight hours at the specific geographic location (lon, lat) (Hamon-PET; Hamon, 1961). The actual ET consists of evaporation terms from both interception and soil moisture storages (see Viviroli et al., 2009)."

L139 what does "satisfactorily" mean in terms of performance, please specify

Thank you for pointing out this imprecise statement which is related to your previous suggestion of providing more detailed information on model calibration and validation statistics. We now circumvent the use of "satisfactorily" by directly referring to the Table B1 in Appendix B (see Table 2) containing calibration and validation metrics representative for low flow situations ($NSE_{log}$ and $KGE_{log}$).

Line 139 [149–163] → see 2. in previous corrections
Lines 458–463 [480–490] → see 3. in previous corrections

L441 Section ???

The section referencing was updated/corrected (see also *Additional adjustments*).

**Review 2**

**Comment on nhess-2024-6 (Anonymous Referee #2)**

Referee comment on "Compound droughts under climate change in Switzerland" (nhess-2024-6) by von Matt et al., Nat. Hazards Earth Syst. Sci. Discuss., https://doi.org/10.5194/nhess-2024-6-RC2

Revision of the manuscript number "nhess-2024-6" entitled "Compound droughts under climate change in Switzerland".

This manuscript contributes to analyzing compound droughts in different catchments in Switzerland under two circumstances: modelling present and forecasting future climates. The paper is well-structured and written overall and can be published in its present form.

Thank you very much!

I have a few technical comments:

1. L28. Change redaction.

   We adjusted the redaction (newline was removed) and slightly rephrased the corresponding sentence.

   Lines 28–29 [28–29]: "There is no single definition of droughts that covers all aspects of the drought phenomenon (Wilwhite and Glantz, 1985; Lloyd- Hughes, 2014; Van Loon, 2015; Brunner et al., 2021; Ault, 2020)."

2. L54. What do you mean by "strongly non-linear"? Be more specific with the degree or type of the function, or just mention it as "non-linear".

   Thank you for pointing out this imprecise statement. We now adjusted the sentence to be more generalizing:

   Lines 52–55 [52–55]: "The exact sequence of the drought signal trans- lation through the hydro-terrestrial system may differ depending on drought typology, drought generating processes, and on human interactions (e.g., water abstractions) and is often non-linear in nature (Brunner et al., 2023; Haile et al., 2020; Savelli et al., 2022; Tijdeman et al., 2018; Van Loon, 2015; Van Loon and Van Lanen, 2012)."

3. L55. Avoid using qualifiers such as "strongly", instead, be more specific with the type of relationship between the variables.

   Thank you for this suggestion. In this specific sentence we excluded the qualifier "strongly". The sentence is now as follows:

   Lines 55–59 [55–59]: "While meteorological droughts are tied to climate vari- ability (precipitation), soil moisture and hydrological drought characteristics are spatio-temporally more variable due to the importance of local factors such as water storage and release or catchment characteristics (e.g., Apurv et al., 2017; Apurv and Cai, 2020; Denissen et al., 2020; Haslinger et al., 2014; Peña-Angulo et al., 2022; Staudinger et al., 2017, 2014; Sutanto and Van Lanen, 2022; Tijdeman et al., 2018; Van Lanen et al., 2013)."

4. L103-L106. Improve redaction.

   Similar to the first suggestion, the extensive spacing was removed.

5. L119-L120. It could be better if you slightly describe what you show in every single section, not only writing the section title.

Thank you for your suggestion. We complemented the most important sections with additional information on section contents. The paragraph is now as follows:

Lines 119-120 [119–126]: "The remainder of this paper is structured as follows: In section *2 Data* the catchments and (model) simulations are introduced. In section *3 Methods* drought indices are presented, the compound events are defined and the climate change assessment approach is described. Section *4 Multivariate compound droughts* presents the results for compound droughts on catchment-level (aggregated on Greater regions) while section *5 Spatial extent of multivariate compound droughts (Spatially compounding droughts)* presents results related to the spatial extent of compound droughts (across multiple catchments). In section *6 Discussion* the results from previous sections are wrapped up with a discussion on plausibility and uncertainties inherent to the present analysis and section *7 Conclusions* then concludes with the most important findings and future prospects in terms of mitigation and adaptation actions."

6. L123. Fix units. 1702 km2 and 1500 m.a.s.l.

The units were fixed from km2 to $km^2$ and m asl to m.a.s.l. Other instances have been checked too (see *Additional adjustments*).

7. Figure 7, the label on the ordinate might be better if it says "probability".

We adjusted the label according to your suggestion from 'Density' to 'Probability'. Further, the color-scale of the figure was adjusted following suggestions from the editor (see *Additional adjustments*).

**Additional adjustments**

**Suggestions from previous editor and consistency adjustments**

In this section, additional corrections which were suggested by the (previous) editor are listed. Note: The assigned editor for this manuscript has changed since.
Further, several minor adjustments mainly considering grammatical inconsistencies or adjustments to *NHESS* house standards were made (see also *Tracked Changes* (Latexdiff)).

The changes are as follows:

1. Line 354 [377]: Incorrect subsubsectioning. Seasonality was changed from a subsubsection () to a regular subsection (4.4 Seasonality).

2. Zenodo Repository was updated by incorporating tables on catchment-climate characteristics and model calibration/validation statistics.
   → see `https://doi.org/10.5281/zenodo.10908410`

3. Figure 7: Caption of Fig. 7 included a description on uncertainty bounds which were not shown in the figure to enhance readability/clearness. The sentence was therefore deleted.

   The adjustments are as follows:
   "Seasonality of compound drought days. Shown is the median value aggregated over all catchments per Greater regions and day of year (DOY) (lines). Probability density distributions are shown for both the mitigation (RCP2.6) and the non-mitigation (RCP8.5) scenario for the periods reference, 2035, 2060, 2085 (colored). Dashed lines indicate the extended summer season from (begin of) May (DOY 122) to (the end of) October (DOY 305)."

4. Several inconsistencies were adjusted and *NHESS* standards adopted, including (among others):

   1) en–dashes, 2) change of "Southern Switzerland" to "southern Switzerland".

   → See also *Tracked Changes* (Latexdiff).

5. Color-scales were adjusted towards colorblind-friendliness following the suggestion of the (previous) editor.

   The following Figures have been adjusted in the manuscript:

   Figure 1, Figure 7, Figure 8 and Figure 9

   In the *Supplementary material*:

   Figure S1, Figure S2 and Figure S3

[revised manuscript text omitted]

[a]The metrics $NSE_{log}$ and $KGE_{log}$ are indicative of the representation of low flow conditions in the Hydro-CH2018 hydrological model simulations (see Muelchi et al., 2022). Both metrics range from $[-\infty$ to 1], with 1 equal to a perfect performance and values $> 0$ equaling to a better predictive performance than the mean of observations. The calibration and validation periods cover period 1985–2014 for most catchments. Even years were used for calibration and uneven years for validation. For more information see Muelchi et al. (2022).